# Hospital-at-home care in Singapore: A qualitative exploration of health system partners' state of readiness, and policy and implementation strategies essential to support scale-up

Crystal Min Siu Chua[1,2]*, Eward Wei Zheng Lim[1]*, Win Hon See Tho[1], Yuka Asada[3], Karen E. Peters[3], Yi Feng Lai[1,2,3]

**1** MOH Office for Healthcare Transformation, Singapore, Singapore, **2** School of Public Health, University of Illinois Chicago, Chicago, Illinois, United States of America, **3** National University of Singapore, Singapore, Singapore

* crystal.chua@moht.com.sg (CC); e0727136@u.nus.edu (WL); seethowinhon@u.nus.edu (WS); yasada2@uic.edu (YA); kpeters@uic.edu (KP); yifeng.lai@moht.com.sg (YL).

## Abstract

### Rationale

Many Hospital-at-Home (HaH) programs have proliferated in recent years to cope with the increasing demands of an ageing population and global hospital bed shortages. Singapore has implemented its own version, Mobile Inpatient Care at Home (MIC@Home). However, many HaH programs remain small, raising concerns about their scalability. Hence, a clear implementation strategy is needed.

### Objectives

To address: (1) What is the readiness of Singapore's health system partners to scale up MIC@Home? and (2) What multi-level strategies are necessary for the successful scaling of MIC@Home in Singapore?.

### Methods

A descriptive qualitative study design was used. Through purposive sampling, 32 participants (16 HaH clinicians, 11 enabling units, and 5 regulators) were recruited and semi-structured interviews were conducted. The interviews were transcribed using Trint and thematically analysed using Atlas.ti via Braun and Clarke's six-step inductive approach. This analysis was guided by the Health Policy and Partnership Readiness Assessment Framework.

### Results

The key themes were: (1) perceived readiness to scale, focusing on stakeholder motivation and capacity; (2) implementation strategies, highlighting the need for

**Data availability statement:** All relevant data are within the manuscript and its Supporting Information files as Supplementary Files 1, 2, and 3. We would like to confirm that our minimal data set is present. I confirm that the manuscript and supporting files (S1 Material. jpg, S2 Interview Guide.docx, and S3 Material. pdf) include the minimal data set as defined by PLOS ONE. No further data uploads are required.

**Funding:** The author(s) received no specific funding for this work.

**Competing interests:** The authors have declared that no competing interests exist.

training, collaborations, and operational refinements; and (3) policy strategies, addressing financial sustainability, governance, and regulation. MIC@Home is seen as a viable solution to acute bed shortages, with high readiness for scaling. Effective governance requires stakeholder buy-in, organizational alignment, partnerships, and adequate manpower. Regulatory strategies should be adjusted to sustain MIC@ Home and improve patient access. For service provision, standardized guideline and data is vital to prove MIC@Home's effectiveness and safety, while convincing clinicians and patients of its value will increase acceptability. Finally, refining governance and establishing regulations for minimum care standards will support smooth operations and long-term success.

## Conclusion

Despite the challenges of scaling MIC@Home, the findings underscore the potential of MIC@Home to enhance healthcare delivery through identifying readiness and strategies to position MIC@Home as an alternative to traditional care.

## Introduction

For centuries, the conventional hospital care paradigm—characterized by patient admission and extended stays in assigned beds until recovery—has been the standard practice [1]. However, the acute shortage of hospital beds, exacerbated by the COVID-19 pandemic, an aging population, and the rise in complex chronic conditions, has put significant strains on acute hospitals [2,3]. In response, Hospital-at-Home (HaH) programs have emerged as a promising alternative. By definition, HaH programs offer active treatment by healthcare professionals, including doctors, in the patient's home for conditions that would otherwise require acute hospital inpatient care, but only for a limited period [4].

A meta-analysis of 20 studies by Edgar and colleagues demonstrated that HaH delivers clinical outcomes and self-reported health status comparable to those of conventional hospital care [4]. Additionally, a qualitative evidence synthesis of 52 studies by Wallis and colleagues consolidated the perceptions of HaH among patients, clinicians, and caregivers, revealing that many stakeholders believe HaH helps achieve favorable health outcomes and high patient satisfaction [5]. Similarly in Singapore, Mobile In-patient Care (MIC@Home), the HaH equivalent, has been shown to be safe and feasible alternative [6]. Prior study has also studied the perceptions of patients and caregivers in Singapore, and found high acceptance rates among these stakeholders [7].

Although promising, the program continues to face challenges in gaining widespread adoption [8]. For example, a Cochrane editorial has highlighted that only about 89 home beds per day were used for HaH across the USA, despite more than 300 hospitals across 37 states having been issued the Acute Care at Home Waiver [9]. Furthermore, Wallis and colleagues' synthesis found that few studies investigated implementation, and none considered the perspective of system-level stakeholders on the scalability of HaH programs [5]. Similarly, in Singapore, the scalability of such

programs on a national level is still uncertain, with existing pilot projects providing insufficient data to guide best practices for expanding home-based hospital care [7].

The absence of a clear strategy for scaling up these programs marks a significant gap in knowledge, contributing to healthcare providers' hesitancy in adopting this care model more broadly [10]. Addressing this gap requires qualitative research in identifying key factors that enhance readiness, implementation, and policy strategies, enabling the successful expansion of hospital-at-home care and allowing more patients to benefit from this innovative approach [11].

Walker (2020) proposed the R = MC² heuristic to measure organizational readiness for implementation, considering both motivation and capacity factors [12]. Readiness has been defined in various ways, such as by Greenhalgh et al. (2004) as the "state of preparation" to accept and implement innovations [13], and by Damschroder et al. (2009) as the psychological and behavioral preparedness of organizational members [14]. Several sub-constructs of readiness have been identified, including staff, organizational, intervention, and leadership readiness (Damschroder et al., 2009), as well as others like resources, patient engagement, and data infrastructure [15,16]. For this study, the following sub-constructs relevant to MIC@Home scaling—Motivation, Capacity, Internal Needs, External Demands, Barriers, and Facilitators— were adopted, and the topic guide was designed to address these constructs [17]. The Health Policy Partnership's Readiness Assessment Framework captures this multidimensional nature of readiness and will be used in the discussion to assess readiness qualitatively [18]. This framework was chosen because it provides a structured approach to assess key enablers and barriers at different levels of the health system. This framework allows for a comprehensive evaluation of stakeholder commitment, system capacity, and policy alignment—factors critical to scaling MIC@Home. By using this framework, the study ensures a holistic and qualitative assessment of Singapore's health system partners, identifying the necessary strategies to facilitate successful implementation and expansion. Overall, this study aims to address two pivotal questions: [1] What is the readiness of Singapore's health system partners to scale up the Mobile Inpatient Care at Home (MIC@Home) care model? and [2] What multi-level strategies are necessary for the successful scaling of MIC@Home in Singapore?

By focusing on the preparedness of health system partners and identifying the strategies needed for the model's adoption and growth, this study seeks to delineate the patterns, obstacles, and facilitators for the implementation and scalability of HaH.

## Methods

### Study Design and Sampling

This descriptive exploratory study used in-depth semi-structured individual interviews to understand the stakeholders' perceptions on the readiness, challenges, and enablers for scaling and mainstreaming HaH across Singapore. Thirty-two stakeholders, including officials from the Singapore Ministry of Health (MOH) and representatives from seven public healthcare institutions across central, eastern, and western Singapore, were purposefully sampled. The interviews were conducted from September to October 2023. This strategy allowed us to capture factors influencing the implementation readiness and scalability of HaH, also known locally as MIC@Home.

The inclusion criteria were relevant key informants who had influence in directing MIC@Home implementation. They have been categorized into three stakeholder groups for a holistic perspective on the MIC@Home implementation, 1) regulators, 2) enabling units, and 3) clinicians. Patients and caregivers have been excluded from this study as they have previously been interviewed [7].

The first category comprised regulatory stakeholders, notably the MOH and its policy divisions. Their decisions are important in shaping the regulatory environmental of new healthcare models like MIC@Home, making it crucial to align the model's strategies with existing guidelines [19].

The second category of stakeholders included enabling units, such as healthcare administrators capable of facilitating the implementation and scaling up of the MIC@Home model. Their role involves integrating the model within existing

healthcare systems, providing resources like information technology support, clinical resources, and data management to optimize operational efficiency.

Lastly, clinicians and implementation stakeholders, directly involved in executing the MIC@Home model, focused on the operational challenges and opportunities during the implementation process.

## Setting

The MIC@Home care model is Singapore's HaH equivalent and was piloted in 2019 at two major acute hospitals in the northern and western regions of Singapore. Till date, this model has been adopted by eight public hospitals across the country, encompassing three academic medical centers, four general hospitals, and one hospital specializing in pediatrics and women's health. Patient referrals came through Early-Supported Discharge (ESD) for patients transitioning from hospital wards, and Admission Avoidance (AA) directly from emergency departments, outpatient clinics, or primary care settings. The patient eligibility criteria for MIC@Home, were set by individual hospitals and were not limited to diagnoses. Therefore, patient eligibility criteria may have differed across hospitals, but targeted patients with acute, general medicine care needs as the guiding principle. Common diagnoses included cellulitis, urinary tract infections, viral diseases, rhabdomyolysis and fluid overload.

Once enrolled, patients were remotely monitored and managed by care teams, with home visits scheduled up to three times a day as needed. Patients stayed in the MIC@Home program until they met discharge criteria. If a patient's condition worsened, physicians could escalate care or arrange for hospital admission.

The theory of change for MIC@Home is built around the premise that delivering hospital-level care in the home environment can improve patient outcomes while reducing strain on hospital resources [7]. MIC@Home core principles include technology-enabled care, community-anchored hospital care, reduced overall healthcare cost, shared care, and providing patient-centered care (S1 Material). The model assumes that, when key conditions are met—such as appropriate patient selection, effective integration of technology, and strong partnerships—MIC@Home can enhance healthcare delivery by offering high-quality, sustainable care.

A previous local study has found that pilots of MIC@Home were well-perceived by participating patients and caregivers in terms of improved clinical outcomes and enhanced quality of life [7]. However, such benefits and their impact on reducing bed occupancy can be further materialized with implementation and scaling-up strategies [9]. Building on this existing evidence, the present study aims to identify potential strategies for MIC@Home upscaling to amplify its outcomes on hospitals. Additionally, this study's research questions are technical and require a deep understanding of the healthcare system and service delivery models. As most patients and service users lack this level of insight, we focused our sampling on the specified groups who could provide relevant expertise.

## Data Collection and Analysis

Participants were recruited via email, and interviews were conducted and recorded over Zoom. Participants were de-identified with the prefix IR, representing implementation research, and a two-digit number (e.g., IR02). Four interviewers participated: Two experienced qualitative researchers (YFL and CMSC), and two pharmacy students (WZEL and WHST) supervised by CMSC. Although YFL knew some participants, all interviewers remained objective and avoided influencing responses.

An interview guide was constructed based on literature [20], crafted under the administrative expertise of one of the interviewers (YFL) and another clinical expert (SK) who was not involved in this study. This was then piloted with one participant. Since no changes were made to the interview guide (S2 Material), the piloted interview was also included in the final analysis.

Interviews were conducted in English, and recordings were subsequently transcribed using Trint - an AI-assisted transcription software, then coded by the researchers using Atlas.Ti software.

A Braun and Clark's six-step inductive approach was used in the thematic analysis process which included: [1] familiarizing with data, [2] forming initial codes, [3] exploring potential themes, [4] reviewing themes, [5] defining and naming themes, and [6] finalizing the findings. Three researchers (CMSC, WZEL, and WHST) analyzed the data and discussed extensively to identify the themes. Any discrepancies were discussed and resolved via the fourth researcher (YFL). The Health Policy Partnership's Readiness Assessment Framework guided the theme and subtheme development, ensuring a structured yet contextually relevant analysis by capturing key enablers and barriers at different levels of the health system and identifying strategies for successful implementation and scale-up.

### Reflexivity and Positionality Statement

All authors are well-versed in public health, working and studying in tertiary academia or public healthcare institutions. CMSC has a background in nursing studies, including the completion of a Master of Science in Nursing. WZEL and WHST are students of a Bachelor of Pharmacy (Hons.) program, who actively participated in research projects centered on MIC@Home in Singapore, contributing to efforts in its implementation. Given their backgrounds and involvement in patient care and healthcare, they have familiarity with cultural aspects and challenges within Singapore's healthcare system with influence on MIC@Home implementation, which may have informed their interpretation of the results. Overall, their reflexivity and positionality reflect their collective commitment to presenting an objective view in this exploration of the care model that could bring significant benefits to both patient and provider when more widely adopted. None of them have direct relations with the interviewees.

### Ethics Approval

Written consent was taken from all study participants and the study adhered to rigorous ethical standards, with protocol approval from the National University of Singapore's Institutional Review Board (NUS-IRB-2023–543) in August 2023. All participants consented, and this study is guided by the Consolidated Criteria for Reporting Qualitative Research Checklist (COREQ) [21] as guideline (S3 Material). Before the interview, interviewees were informed that the interview is anonymous, that they can withdraw anytime, and that participation is voluntary.

## Results

### Participant Characteristics

Thirty-nine potential participants were approached from September to October 2023, where 32 agreed to participate, and seven declined due to conflicting schedules. The demographics of the 32 participants are shown in Table 1. On average, the interviews lasted 55 minutes.

The results are organized into three themes, as depicted by the colored oval shapes in Fig 1. These included [1] the perceived readiness to scale, focusing on stakeholder motivation and capacity; [2] the implementation strategies, highlighting the need for training, collaborations, and operational refinements; and [3] the policy strategies, addressing financial sustainability, governance, and regulation. Each theme was divided into different aspects known as subthemes, shown by the colored rounded rectangular shapes in Fig 1. The subthemes would then branch out into different topics that were discussed during the interviews, which are seen in Figs 2–5. These findings underscored the multifaceted approach required for the effective scale-up of the MIC@Home care model, integrating organizational, technological, and human resource considerations to meet internal and external healthcare demands.

### Readiness to Scale

This theme aimed to address objective one – "What is the readiness of Singapore's health system partners to scale up MIC@Home?" – through stakeholders' motivation, perceived internal needs and external demands. Motivation to scale

**Table 1. Characteristics of Interviewees.**

| Key informant category | Qualitative Interview Participants (n = 32) |
|---|---|
| **Clinicians** | |
| Doctor | 10 (31%) |
| Nurse | 3 (9%) |
| Pharmacist | 3 (9%) |
| **Enabling Units** | |
| Healthcare Administrator | 4 (13%) |
| C-suites | 4 (13%) |
| External Healthcare Advisor | 1 (3%) |
| **Regulators** | |
| MOH Stakeholders | 7 (22%) |

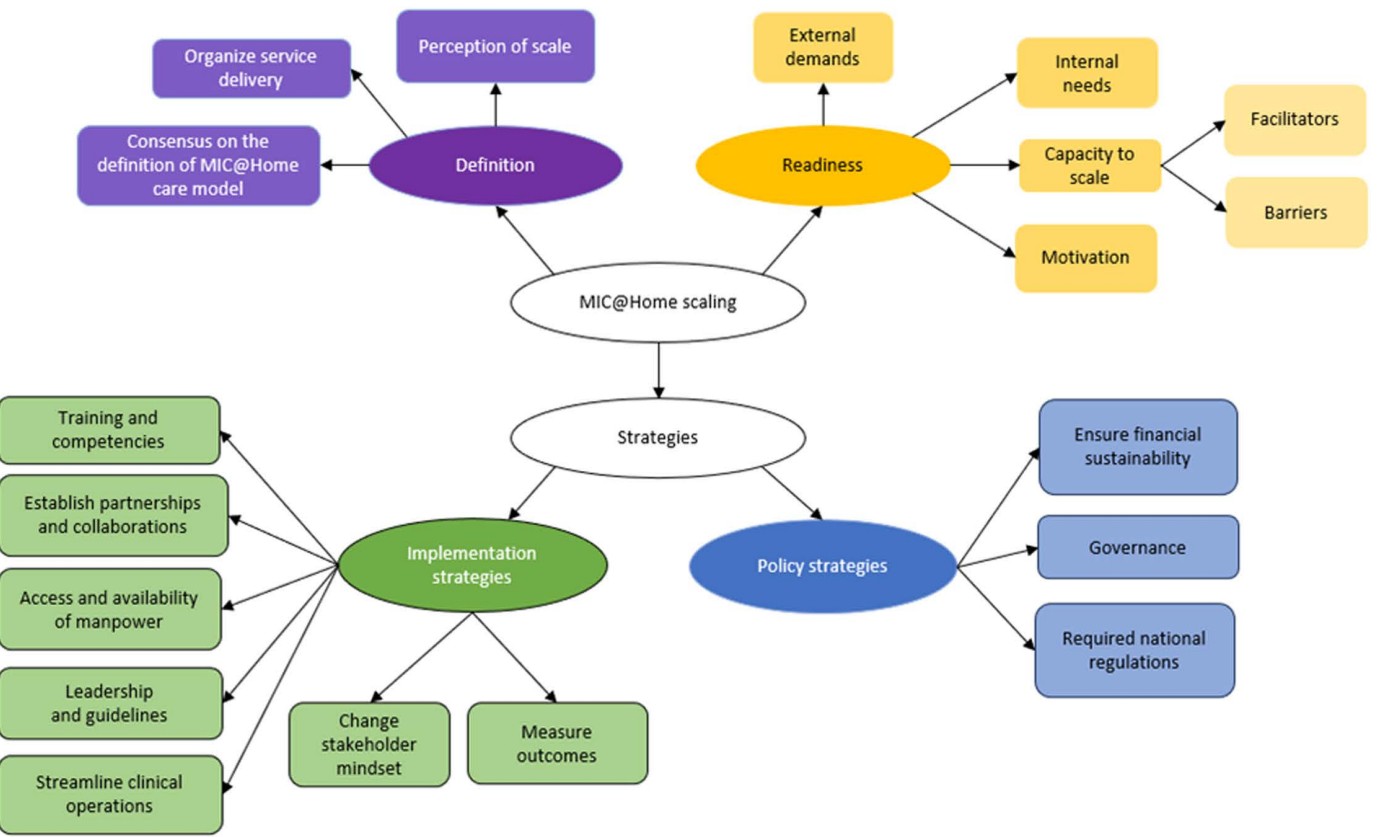

**Fig 1. Overview of Interview findings.**

stemmed from the prospect of improved patient outcomes and satisfaction. Internal needs reflected the drive to optimize care right-siting and cost-effectiveness. External demands were centered around solving bed capacity issues. Capacity to scale was categorized into facilitators and barriers, aligning with the Health Policy Partnership's Readiness Assessment Framework's focus on stakeholder commitment, system capacity, and policy alignment. Perceived barriers—such as stakeholder buy-in, clinician manpower constraints, and IT infrastructure challenges—reflected gaps in system readiness.

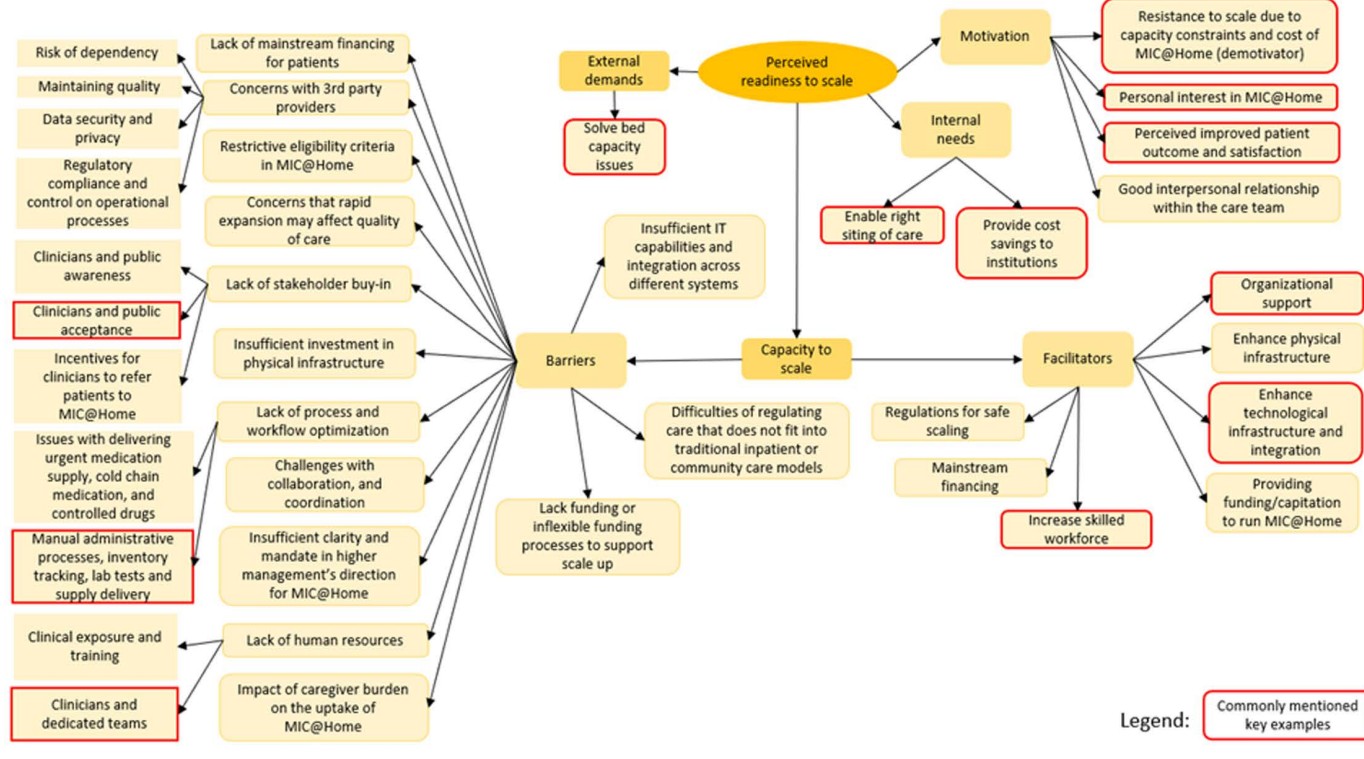

*Figure 2: Perception of stakeholders' readiness to scale MIC@Home*

**Fig 2. Perception of stakeholders' readiness to scale MIC@Home.**

Conversely, facilitators—including organizational backing, enhancements in physical and digital infrastructure, and workforce development—highlighted key enablers supporting scale-up efforts. These findings were mapped onto the framework's subconstructs to ensure a structured and comprehensive understanding of readiness (Fig 2).

**Motivation.** The data suggested that the perception of improved care quality, patient outcomes and satisfaction was a significant motivator for scaling the HaH model.

*"...the motivating force - **we can see that there's a great impact on the patients...I can say 95% of them have given very good feedback**…**without this care model, there'll be no other alternative for them to achieve care at home**...this is what the is the key motivator..."* (IR03, Clinician)

Some participants expressed a personal interest in MIC@Home, and this could drive thesuccessful scaling of HaH.

*"….the meaningful work that…I'm currently in engaged in right now for **MIC@Home [is] to resonate with what I want for myself** when I get old."* (IR12, Clinician)

Each of these findings indicates the complex interplay of motivators in the readiness for scaling MIC@Home services. In alignment with the Health Policy Partnership's Readiness Assessment Framework, fostering readiness can drive sustainable MIC@Home implementation and improved population health outcomes.

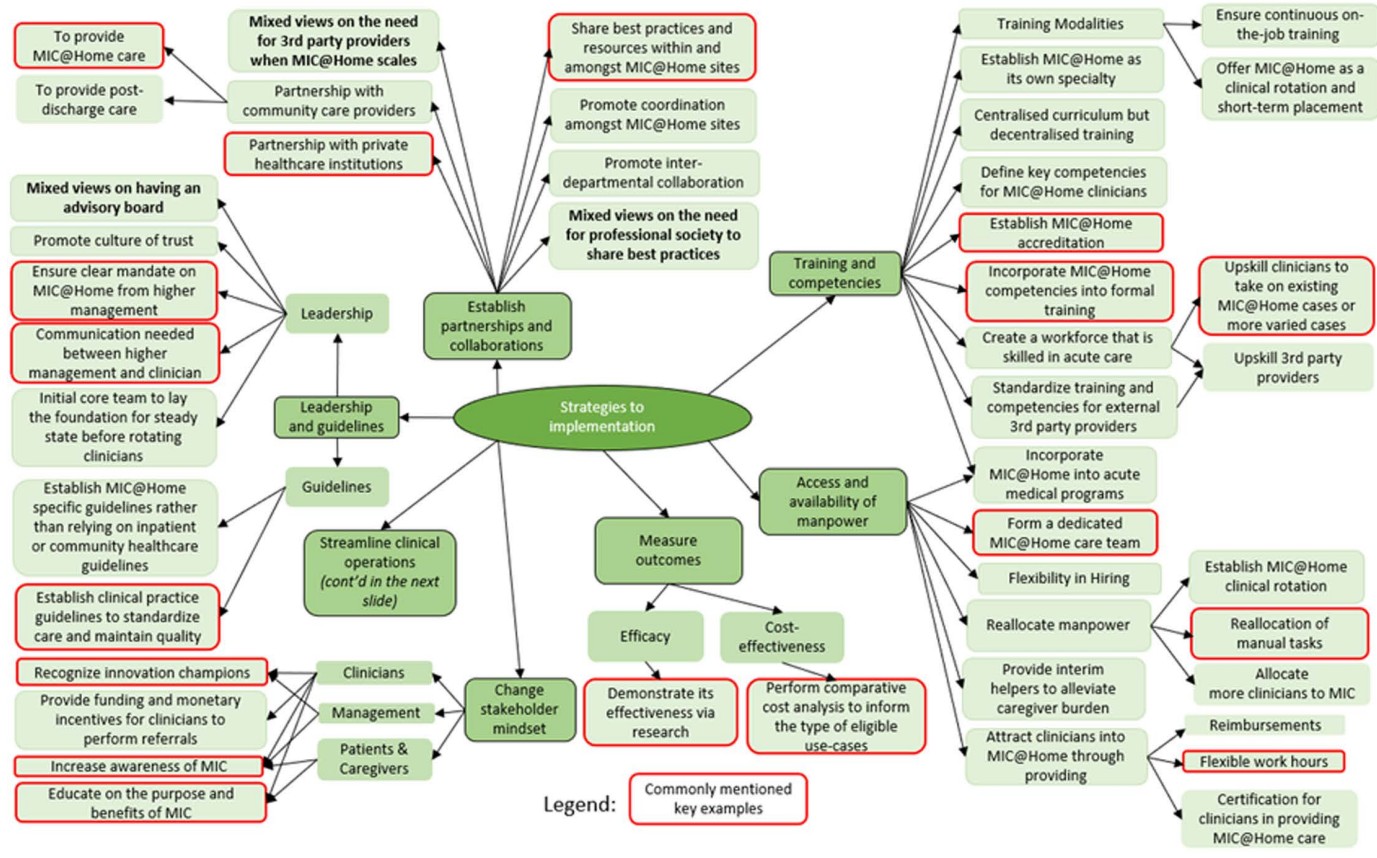

**Fig 3. Implementation strategies uncovered through qualitative interviews.**

**Internal Needs.** Two principal findings emerged concerning the inherent demands driving the push for MIC@Home: [1] Enabling right-siting of care, and [2] Operational cost-savings.

Participants see MIC@Home as a strategic approach to ensure patients receive the right level of care in the mostpatient-centered setting. This is viewed as a critical component of enhancing healthcare delivery and patient experience.

*"I think from clinicians' point of view…is that it's also right siting…you also **recognize the benefits of treatment at home**, right? Without coming to a hospital, **less nosocomial infections, less problems, less force and delirium for older patients..."** (IR05, Clinician)*

While some stakeholders are doubtful about the cost-savings, there are several others who believe HaH can reduce healthcare system costs.

*"…not talking about the CAPEX [capital expenditure] savings, there may be **a 20% cost savings from this model. I mean, that got us very interested..."** (IR27, Regulator)*

These insights reveal that both optimization of patient care and financial considerations are key drivers for MIC@Home initiatives.

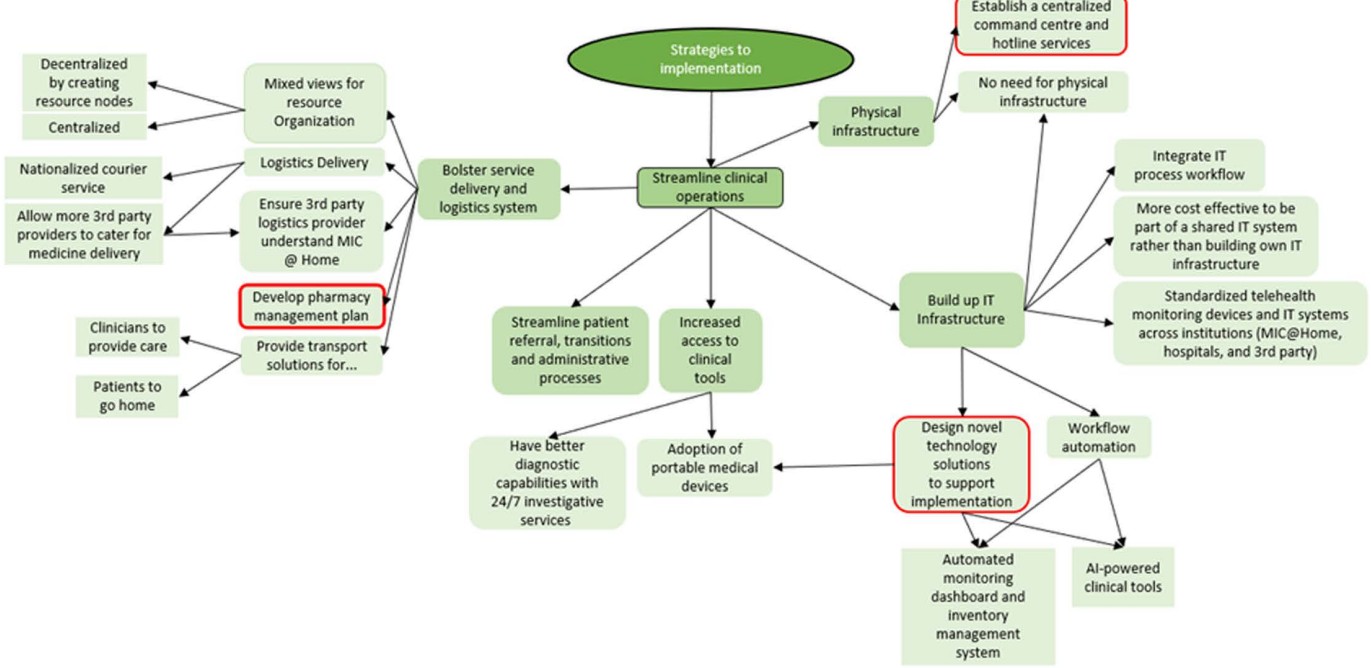

**Fig 4. Implementation strategies uncovered through qualitative interviews.**

**External Demands.** The key findings around external demands are related to addressing acute bed demands and capacity challenges. Participants identified MIC@Home as a potential solution to alleviate hospital bed shortages.

*"So, [there are] **plans to add capacity**, [a public hospital] and maybe another hospital in the west [of Singapore] later. Then one other event that is going to take place also is [a public hospital] renovation or redevelopment of [a public hospital]. So, there'll be a period as well where the beds would be reduced."* (IR14, Healthcare administrator)

**Barriers to Scaling-up.** Besides positive motivators, the findings delineated three primary barriers: [1] Lack of health system partners' buy-in, [2] Insufficient human resources, and [3] Process and workflow optimization Issues.

The study uncovered reluctance among clinicians to adopt the MIC@Home model due to unfamiliarity and consequent hesitation to refer patients.

*"...the large majority [of clinicians] **have not experienced** what MIC@Home is or what it can do for their patients. And therefore, **most clinicians would be hesitant...to come on board or refer their patients."*** (IR04, Clinician)

The findings also highlighted the challenge of allocating additional or redirecting existing staff to support MIC@Home, with an emphasis on the scarcity of clinician resources.

*"So you must always **prove that you deserve…more manpower**…But it's a vicious cycle. **If you don't have enough manpower, you can't really get more patients in.**"* (IR08, Clinician)

Current manual administrative processes were identified as barriers to efficient referrals, though there were plans to integrate these into the electronic medical record system for a more streamlined experience.

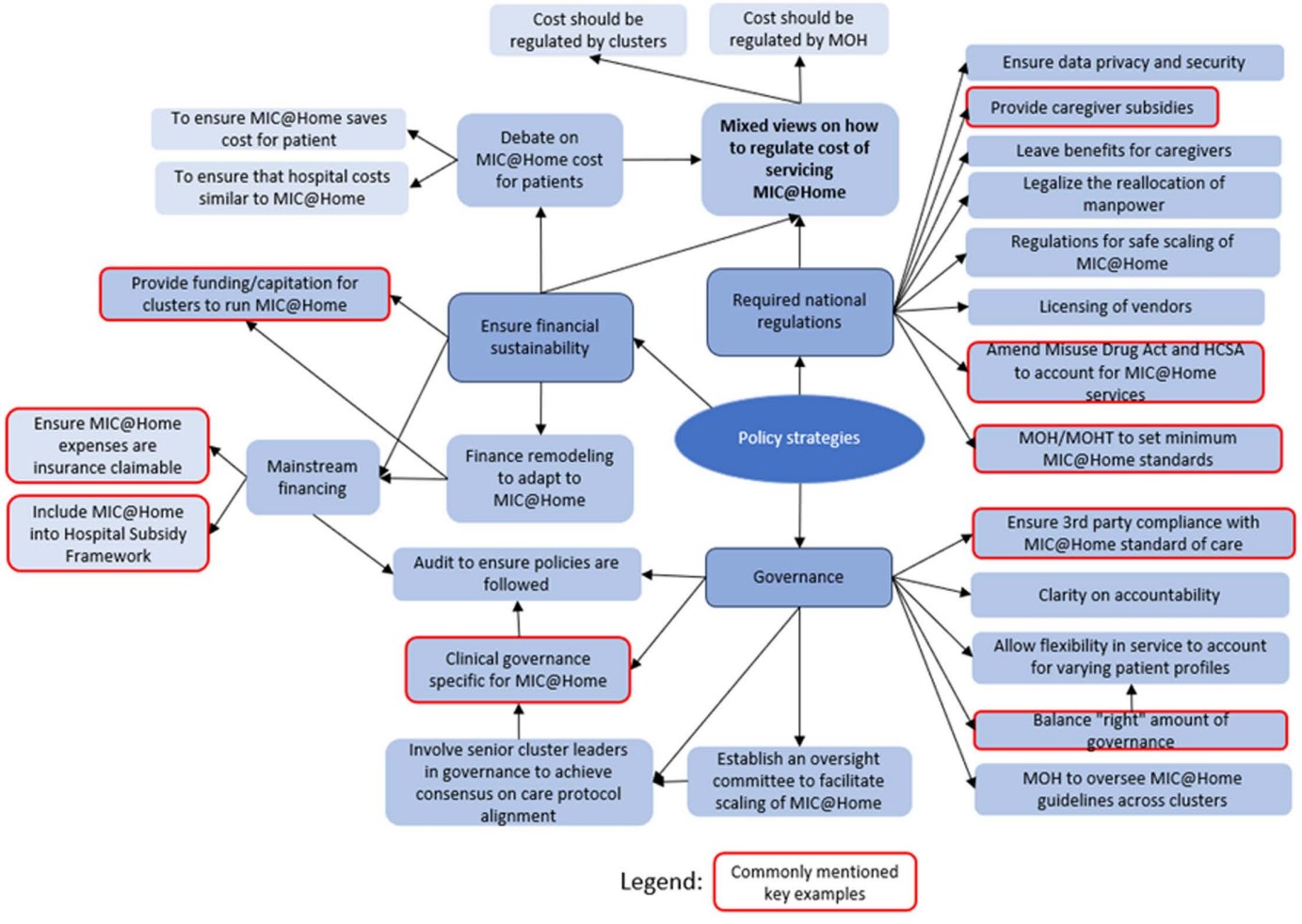

**Fig 5. Policy strategies uncovered through qualitative interviews.**

*"...there should be EMR (electronic medical record) integration. I mean, **not just the IT aspect, but for the other ancillary services** etc. And I guess assimilation into the BAU (business as usual) systems becausea lot of the issues come from it being part of the sandbox... And hence **a lot of additional... administrative work…**"* (IR19, Enabling Unit)

These barriers underscore the complexities in expanding HaH models, pointing to the need for increased awareness, resource planning, and process improvement.

**Facilitators to scaling-up.** The results highlighted three elements that promoted the expansion of MIC@Home: [1] Organizational support, [2] Technology adoption, and [3] Availability of skilled workforce.

Strong backing from senior management, particularly department heads and chiefs, was essential in gaining support and approval for collaborative efforts.

*"Our intervention or collaboration have been **highly supported by our senior management** because whenever we have any collaboration with other department, you know, our **HOD will be the one driving** this...without their approval we cannot have this conversation."* (IR16, Enabling Unit)

The need for unified digital platforms was evident. Current practices involve a mix of communication tools, and there was a move towards creating a harmonized infrastructure to streamline processes.

*"I think **we probably need a more unified or harmonized way of doing things**…**we are using a hotchpotch of vital signs monitoring**, we use Zoom sometimes, or WhatsApp video if the Zoom fails, so it's not ideal…**And I believe we're working on that sort of a digital infrastructure to streamline the processes...**"* (IR10, Clinician)

These facilitators were seen as key to successfully scaling up the HaH model, indicating the importance of leadership, technology, and workforce development.

**Implementation Strategies to Enable Scaling Up.**

This theme aimed to answer objective two – "What multi-level strategies are necessary for the successful scaling of MIC@Home in Singapore?". By integrating insights from interviews and the Health Policy Partnership's Readiness Assessment Framework, the analysis identified key implementation strategies: leadership through organizational alignment, revamping guidelines, building partnerships and collaborations, strengthening workforce capacity, measuring outcomes, shifting stakeholder mindsets, and enhancing operational efficiency (Fig 3 and 4). Mapping these strategies to the framework ensured a structured assessment, linking them to critical readiness dimensions and capturing the specific strategies essential for scaling MIC@Home.

**Leadership & Guidelines.** Two key ideas emerged regarding the importance of leadership, namely that [1] Higher management should have a clear mandate and [2] The need for higher management to communicate with the clinicians.

Higher management should establish a clear directive for MIC@Home, so that the ground can make changes to suit the needs of the program.

*"I think **the directive should come from the senior management** to say that we need to build MIC and, and please work with the team to see what is required to enable this model- to scale up…"* (IR15, Enabling Unit)

Some also highlighted the importance of clear communication between the clinicians, institution-level management, and the ministry to ensure alignment of MIC@Home goals and consistency in future steps.

*"I wonder if **the higher management should also be looped into this discussion** because I think they may not be aware of where we are heading towards…So there's a misalignment and when it comes from ground up to reach them, it's quite difficult."* (IR06, Clinician)

Another recurring point mentioned by various interviewees is that clinical practice guidelines should be standardized to maintain quality of care and establish clearer scopes of practice for clinicians.

*"...to do some playbook so that **we can standardize services across hospital**, providing MIC@home care and more of the national policies. So **those supporting documents will help existing MIC@homesadhere to certain preset policies...**"* (IR13, Clinician)

**Establish partnerships and collaborations.** Collaboration was an idea that was shared, encompassing the need to establish a robust network through: [1] Partnerships with community care providers to provide MIC@Home care, [2] Partnerships with private healthcare institutions, [3] Sharing best practices and resources across different MIC sites.

Given the decentralized nature of MIC@Home, having the patients all situated in their own homes, some cited the possibility of tapping into community-based resources like polyclinics to lighten the load on the PHIs.

*"... why don't we tap on primary care? **There are many polyclinics around right?** What if we engage the polyclinics to help us with some of these things?... **They might be closer to the patients so can we tap on that**?"* (IR08, Clinician)

Furthermore, as the MIC@Home program expands, the increase in capacity likely needs to be shared with the private sector, including private healthcare institutions.

*"It will **require a joint partnership between the public and private to scale it up**, because unfortunately, I think that our public healthcare sector is already facing a resource crunch."* (IR26, Regulator)

Sharing of best practices and operation models between various MIC@home sites could expedite the maturing of newer sites and inform certain operational decisions for smoother workflow processes.

*"I like **the idea of clusters in a sense competing because we we get a chance to see different models** as well as you know, oh which partner did you use, and then can I **maybe in the future learn whether you had any issues**."* (IR15, Enabling Unit)

**Training and Competencies.** A few recurring ideas to improve the quality of the healthcare workforce in preparation for this new care model were: [1] Establishing MIC@Home accreditation, [2] Incorporating MIC@Home competencies into formal training, [3] Upskilling clinicians to take on existing or more complex MIC@Home cases.
To ensure clinicians are qualified to provide MIC@Home care, they need to be accredited before being allowed to practice, as with any other specialties.

*"…there can always be add-on modules for training and competency…One way to do this is to have a national standard for MIC@Home... **it's about assessment of care, delivery of care, monitoring and maybe infection control, etc. You must have some standards.**"* (IR35, Enabling Unit)

One way to train clinicians for MIC@Home is to incorporate it into medical school training, and to make it a specialty so that residency programmes can be formalised.

*"I thought maybe the **specialty services, like residencies in their training**, should always think about how you want to deliver care beyond the walls of the hospitals."* (IR35, Enabling Unit)

Capitalizing on the current competencies of the healthcare workforce will be vital in the short- to medium-term, hence upskilling the current clinicians and empowering them to perform MIC@Home care will be necessary.

*"…**my same nurse must be able to manage a colorectal post-surgery patient, a dengue patient, someone who is a cancer patient or someone who might be end-of-life** and going home on compassionate discharge. So, **that same nurse needs to have sufficient knowledge across different conditions, we have to keep upskilling**."* (IR04, Clinician)

**Access and availability of manpower.** Other than improving the quality of manpower, the quantity needed will increase as MIC@Home becomes more widely accepted, through these methods: [1] Form a dedicated MIC@Home care team, [2] Reallocation of manual tasks away from clinicians, [3] Attracting clinicians to join MIC@Home through providing flexible work hours.
Despite the manpower crunch faced by the public sector and therefore lack of manpower to allocate to MIC teams, a fully staffed dedicated team should be assigned to work on the MIC program to scale the program up.

*"…right now, **we have no junior layer** for the doctor. So everything is done by the senior... who has to deal with service development, administrative, workflow development areas…**it does limit the ability to scale**."* (IR06, Clinician)

Certain administrative tasks can be automated or reallocated to auxiliary staff, to reduce the clinician's workload and enable greater focus on clinical care.

*"...if some of these could be worked out then I think that could also help with scaling and also because I think for pharmacy the manpower side is a bit tight and then for nursing-wise, **they have a lot of manual stuff that can be reallocated to like program coordinators or admin roles** that handle the task and then **they can focus more on those necessary tasks**."* (IR19, Enabling Unit)

To attract more clinicians to join the MIC workforce, flexible work arrangements could be a pull factor, to build a more robust repository of manpower to tap on.

*"...you can hire people who are part timers for example. Maybe **they are doctors who have gone part time** and therefore they cannot come back to the hospital system but if you offer them this hospital at home they can work within certain limitations, they will be happy to sign on then you can tap on this extra manpower."* (IR25, Regulator)

**Measure outcomes.** Monitoring MIC@Hom effectiveness will spur further involvement in clinicians as more robust evidence becomes available, and further research can be built to reaffirm its validity.

*"**Support research at the beginning**…And **the best way to do that is not to tell people, but to show people. And the way you do that is by publishing the papers**."* (IR24, Enabling Unit)

On the other hand, measuring cost-effectiveness through comparative cost analysis can demonstrate the financial impact of MIC@Home on society.

*"If it is indeed true that MIC@Home is cost and clinically effective, **we should either be able to see a reduction in the overall costs for the same number of patients**…than building extra hospital capacity."* (IR37, Regulator)

**Change stakeholders' mindsets.** To ensure the program has sufficient enrolment, clinicians and patients must be aware and willing to participate. Key mindset shifts have been identified to do so: [1] Educate on the purpose and benefit of MIC@Home, [2] Increasing awareness of MIC@Home, [3] Giving recognition to innovation champions.
Informing both clinicians and patients on the purpose of MIC@Home will enable them to understand the entire care model more holistically and hence increase acceptability of this program, especially since it is new.
Regarding clinicians

*"…understanding the purpose of MIC@Home and then why the scale...takes quite a fair bit of mindset shift again and…**open communication with the ground [so that they are] engaged and understand why they're doing this**."* (IR12, Clinician)

Regarding patients (and their caregivers)

*"…**educating themon how they can provide this care potentially without disrupting their lifestyles** and their methods. I think **that sort of education might also be useful to incentivize take up among the patients**."* (IR36, Regulator)

Nonetheless, awareness of the program must be made known so patients can consider participating in it.

> *"…**consistent messaging would be key from publicity point of view** that nationallyMOH or MOHT can consider. **Success stories of this, media coverage...**"* (IR12, Clinician)

At the same time, within the MIC@Home sites, there must be clinicians advocating for this new care model (also known as innovation champions), to encourage peers to take this up.

> *"We are **still at the early adopters stage, where we have champions in some of the departments, and to advocate** for this model, be the first ones to test it out. And hopefully, this evidence will then be spread across the hospital."* (IR04, Clinician)

**Streamline clinical operations.** There were many facets optimizing clinical operations, namely fortifying physical and digital infrastructure, increasing accessibility to clinical tools, streamlining referral process to reduce administrative workload, and bolstering service and delivery systems. The more commonly mentioned points were: [1] establishing a centralised command centre and hotline services, [2] designing novel technology solutions, and [3] developing a pharmacy management plan.

Many cited the need for a physical space to coordinate telehealth monitoring and clinical discussions, which correspond to the concept of building a 'command centre'.

> *"The **current PHI has no space for us to expand our command centre so that we can incorporate more staff, to add more equipment to help support more patients.**"* (IR03, Clinician)

Given the rapidly expanding technological landscape, even in healthcare, it would be possible to incorporate novel technologies into the MIC@Home framework to bolster operational efficiency.

Some examples that were brought up were wireless vitals monitoring systems, drones for delivery of medical supplies, digitized financial counselling. Medication management for inpatient care has also been historically difficult due to issues with (a) the frequent medication regimen and (b) the need to titrate or switch medication therapy during the treatment. Hence, a proper medication management plan needs to be developed to ensure optimal treatment outcomes and minimal disruption to patient's daily lives.

> *"As **the more frequent the drug is given, the more drugs get to keep on resupplying** as well. So that would be time consuming both for the pharmacy and nursing side…We are **working out how we can increase the number by [using] infusers.** So now we have a lab pharmacy within [a public hospital] that can help us to do this infuser."* (IR13, Clinician)

**Policy Strategies.**

Policy strategies also aimed to answer objective two – "What multi-level strategies are necessary for the successful scaling of MIC@Home in Singapore?" by ensuring financial sustainability for operating MIC@Home and accessibility for patients, overseeing the maintenance of clinical standards, as well as creating or amending regulations to accommodate MIC@Home (Fig 5).

**Ensure financial sustainability.** Participants shared that there was a need to remodel current financing policies for healthcare institutions, such as [1] to have sufficient capitated budgetto operate sustainably. Additionally, mainstream financing to improve the affordability and accessibility of MIC@Home. Examples of mainstream financing included [2] allowing patients to make use of hospital subsidy such as Medisave, and [3] ensuring MIC@Home expenses are claimable by insurance.

The funding of PHIs in the pilot phase had helped set up MIC@Home and sustained its operations. As such, a mainstreamed MIC@Home would require more funding for its implementation across PHIs in Singapore.

*"...the funding that's been coming from MOHT has been **very helpful in setting up processes**"* (IR10, Clinician)

Since MIC@Home is treated as an inpatient stay at the patient's home, and normal inpatient hospitalisations allows one to use national healthcare subsidy schemes such as Medisave or their own insurance schemes, the same should also be applicable to MIC@Home costs.

*"...[patients] know that if they don't get hospitalized, **they can't use Medisave, they can't use insurance**. And there has always been a major barrier…"* (IR10, Clinician)

**Governance.** As proposed in the Health Policy Partnership's Readiness Assessment Framework, governance is a key aspect of integration to ensure that MIC@Home remains accessible. During the interviews, there were also discussions about ensuring MIC@Home meets appropriate standards through regulatory oversight. The commonly mentioned ideas were [1] to have a clinical governance specific for MIC@Home, [2] ensuring 3rd party vendors adhere to standards, and [3] to have an appropriate balance of governance over MIC@Home.

Due to MIC@Home being a relatively new care model, many participants were unsure about who would oversee and govern MIC@Home, as well as the reasonable boundaries clinicians can practice within.

*"…So, under such a multi-specialty kind of care model, **what would be the clinical governance look like? And who reports to who?**... Any potential safety issues or regulations are well adhered to…"* (IR04, Clinician)

The involvement of 3rd party vendors assisting with MIC@Home operations also raised concerns on complying with hospital standards.

*"...when we engage 3rd parties…the minimum standards… we need to **know the competency level, the qualifications-** are they able to deliver clinical care at the same standards that hospital clinicians can deliver..."* (IR05, Clinician)

Participants also mentioned that there should be an optimal amount of governance over the institutions and other parties.

*"Not to tie up with too many regulations…makes it so **cumbersome in terms of operating and delivering care and also adding costs to the system.**"* (IR34, Regulator)

**Required national regulations.** With plans to mainstream MIC@Home, participants highlighted the need for new regulations, such as [1] setting minimum standards, and [2] subsidizing fees for respite care services. [3] Existing regulations should also be amended to accommodate MIC@Home, one example being the amendment of the Misuse of Drugs Act and Healthcare Services Act (HCSA).

Participants expressed that there should be some form of minimum standards to help guide institutions in delivering patient care with appropriate efficacy and safety.

*"…the **overarching MOH policies and guidelines and structure would just be in place to make sure that the minimum standards are in place** so that patients who are under MIC@home at least they can expect certain levels of care…"* (IR13, Clinician)

Many mentioned the increased stress of primary caregivers of having to take care of patients at home. One way would be to provide subsidies to help hire interim helpers to relieve primary caregiver burden.

*"If…they [MIC@Home caregivers] can get some **subsidies to hire, maybe a temporary help**."* (IR22, Clinician)

Participants have expressed that there is a need for existing regulations to be changed.

The current Misuse of Drugs Act, which governs the use of Control Drugs such as psychotropic substances limits the accessibility of these medications to patients in nursing homes who need it.

*"There's a need to overcome a lot of previous policies that were already set in stone, like things like blood transfusions…Even the **nursing home controlled-drug situation or the Misuse of Drugs Act was written**…"* (IR02, Clinician)

Healthcare Services Act (HCSA) places different healthcare services in different categories, with MIC@Home being categorized as an outpatient service despite being an inpatient care service. Due to the perceived incompatibility of these laws, participants have expressed that these existing regulations be changed to allow for proper MIC@Home service delivery.

*"…there is a bit of incongruence…**we are delivering the inpatient equivalent kind of service but then under HCSA it is parked under outpatient**."* (IR05, Clinician)

## Discussion

To our knowledge, this is the first qualitative study understanding the factors influencing of HaH scale up in Singapore. In this discussion, we categorize the themes from the Results section into four actionable items based on the on the Health Policy and Partnership Readiness Assessment Framework: [1] Governance, [2] Reimbursement and Regulation, [3] Identified Need, [4] Service Provision, and [5] Health Information [18]. Given that MIC@Home has moved beyond its trial phases to scaling operations, the discussion focuses on [1] governance, [2] reimbursement and regulation, and [3] service provision, followed by future steps. Our study underscores the complexity of scaling MIC@Home in Singapore, emphasizing the need for healthcare partners' readiness, effective implementation strategies, and supportive policy frameworks to ensure successful integration into the healthcare system.

### Overall Readiness.

Stakeholders from this study exhibit strong motivation to scale HaH but perceive a lower capacity to do so. Strong motivation was driven by the potential for better patient care, internal needs such as right-siting, and external demands to address bed capacity issues in the hospital. These motivations align with prior global studies on the perceived benefits of HaH [5,22]. However, a significant "de-motivator" was related to doubts about the cost-saving potential of HaH. This is congruent with prior research, highlighting the lack of HaH cost assessments. This hinders the understanding of how and how much healthcare system should invest in scaling HaH programs [23]. While there are growing international studies on HaH cost assessment, a systematic review of reviews noted that these studies were often of low-quality evidence [24]. Thus, a more robust cost analysis, including a country-specific assessment, is needed. Understanding the cost implications of HaH for the country can guide organizations in investing effectively in scaling HaH and build clinician confidence in referring patients to such programs.

Overall, while many expressed high overall motivations to scale, the perceived capacity to scale remains low, plagued by concerns such as inadequate infrastructure, resource shortage, lack of training, and regulatory or legal issues. The following three discussion points will illustrate insights on how ground implementation, policy implementation, and future steps can help alleviate these capacity issues.

### Governance and buy-in is required for mainstreaming MIC@Home.

Effectively administering the HaH model is highly complex, demanding close coordination among medical equipment suppliers, clinical staff, and remote monitoring entities. Therefore, in line with Health Policy and Partnership Readiness

Assessment Framework, strong governance in terms of leadership and planning is important. This includes securing management support, advancing infrastructure, and persuading stakeholders.

Participants from this study spoke of the need to ensure clear mandate on MIC@Home implementation from policy decision makers and hospitals' leadership. Some cited opportunities for additional support in strategic direction and resource provision. It was also noted that clear communication of organizational interests to allow for alignment of strategic goals between the management and ground clinicians would be extremely helpful.

The way to achieve this could be through the engagement of different stakeholders within existing governance structures [25]. This would involve senior cluster leaders' involvement in various platforms such as consultations and forums to achieve a common consensus on issues like workflow alignment. The consultative approach involving various levels of stakeholders can allow for greater communication and hence encourages top-down support to provide the resources required for the scaling of MIC@Home.

This aligns with the service provision domain of the Readiness Assessment Framework which emphasise that workforce capacity should be established. Indicators include determining the roles of each institution or provider involved through proper communication and interaction methods [18].

**Effective communication to garner stakeholder buy-in and public awareness/ acceptance.** There was a lack of awareness among public and clinician colleagues on HaH. To maximize public awareness, it is important to create communication practices that are culturally and linguistically suitable. This should involve effectively conveying the HaH care experience, its benefits, and the procedures for addressing safety concerns [26]. On the providers' end, getting clinician colleagues' buy-in would be crucial to increasing patient referrals. This could be done through innovation champions [23] and engaging clinicians through continuous education, professional development opportunities, and recognition of their role in pioneering new care models. Another possible solution is to make referral easier. For example, the Mayo Clinic's HaH program developed best practice advisories to alert providers of potential HaH candidates, which has helped increase referral rates [27].

**Reimbursement: Important lever to ensure accessibility to and sustainability of MIC@Home.** Many stakeholders have expressed concerns about the lack of subsidies for HaH. Fortunately, since April 2024, HaH services can be funded through Medisave healthcare savings and insurance [28]. This change represents significant progress. However, besides subsidies, findings also highlighted the need to have a suitable financial approach to support the functions of MIC@Home. Discussions from the interviews brought up a lack of funding or inflexible processes to support scale up, the need to have sufficient capitation for clusters to run MIC@Home. Providing adequate financial support for the clusters and healthcare institutions during and after scaling up of MIC@Home may empower sustainability.

As such, innovative funding mechanisms, such as value-based care models could be explored, while tailoring capitation to respective PHI needs, and adopting bundled payments could provide the necessary financial sustainability for HaH programs [29,30]. The Readiness Assessment framework similarly highlights the need for evidence on the suitability and sustainability of such funding mechanisms under its regulation and reimbursement domain [18].

HaH has been shown to place a substantial caregiver burden. Findings suggest that some form of subsidies for respite care may help relieve this. Similarly, other studies have shown that there must be policies in place that minimize the risk of task-shifting, additional expenses to caregivers [26]. Caregivers can be provided financial assistance, implement temporary relief, and offer HaH-related training. Policymakers should also explore including caregiver well-being in the HaH quality metrics.

**Regulation: Robust governance and regulatory oversight to ensure patient safety and sustainable implementation.** There are difficulties in regulating MIC@Home care as it does not typically fit into traditional inpatient or community care models. Changes to existing regulations that were not created to facilitate community-anchored care models, such as MIC@Home, would be needed. Examples included the Misuse Drug Act and HCSA. Revisiting such laws would help to address gaps and limitations in MIC@Home care delivery, as well as to accommodate the unique aspects

of MIC@Home care. The regulation and reimbursement domain of the Readiness Assessment Framework also evaluates existing regulatory policies based on how well-suited or adapted they are to new models of care [18].

On standard setting, participants opined that there was a need for regulators to articulate minimum service standards to all providers. This could be done through establishing guidelines explicitly rather than leaving it to providers to self-regulate. The standards should also apply to all 3rd service providers. Establishing a baseline standard could ensure quality uniformity.

Participants emphasized the need for a balanced regulatory approach for successful HaH implementation. Key points included fostering a culture of trust, establish oversight mechanisms to facilitate the scaling of MIC@Home, allow service flexibility, clarify accountability, and audits. Striking this balance can help avoid operational inconveniences and enable PHIs to refine their own MIC@Home service delivery.

As such, the necessity for robust governance and regulatory oversight cannot be stated. While hospitals are liable for clinical outcomes, policies and regulations that support the safe and effective delivery of HaH services are crucial. Policymakers should work closely with healthcare providers, patients, and technology companies to create a regulatory framework that ensures patient safety, and protects data privacy. This is corroborated by the regulation domain of the Readiness Assessment Framework which states that regulations for the administration of care should be appropriate for safe, effective and streamlined delivery. Moreover, the governance domain of the framework also includes that standardized guidelines with information on how care should be delivered could help to ensure quality and consistency of patient care [18].

**Service Provision: Bolstering infrastructure is crucial to the success of MIC@Home operations.** The need to improve both physical and digital infrastructure was highlighted, especially in areas like medication supply and special transportation. Current guidance on medication management in the HaH model is limited [31]. Therefore, healthcare leaders and local pharmacy boards should create tailored medication management plans, including protocols for storage and delivery [31]. Furthermore, pharmacy support is crucial for the HaH model's success, emphasizing the need for better pharmacy resources to support these care models.

Stakeholders in this study emphasized the need for a centralized monitoring dashboard to track patient status and clinical performance from a single terminal, along with a digital inventory management system for improved efficiency. They also emphasized the importance of standardized telehealth monitoring systems across MIC@Home sites and with 3rd party providers. All of which have a common theme: to enable real-time communication and feedback among patients, caregivers, and the provider team. However, adopting these systems requires financial incentives, capital investments, and regulatory safeguards [26]. As reliance on digital tools increases, policymakers also need to ensure robust cyber-security measures to protect health information from breaches [8]. In addition, tracking patient status and clinical performance calls for the development of localized metrics to provide context-specific estimates and identify equity stratifies (e.g., demographic, social, economic factors). These measures can help identify variations in quality while minimizing the burden on information systems and stakeholders [32].

This is also supported by the service provision domain of the Readiness Assessment Framework which assesses health facility capacity based on appropriate integration and sufficient capacity of supporting healthcare providers to cater to the demand for MIC@Home [18].

### Policy implementation.

Governing or regulatory bodies, such as policy changes and regulatory governance are also required. This involves having adequate human resources and formulating new and amending old finance policies.

**Human resource policy: Adequate quantity and quality of human resource as service backbone.** The findings show a need for a skilled workforce for HaH care. While there has been no increase of malpractice issue for HaH programs [33], it is essential to embed critical competencies as the program scales. Leadership must define the specific

clinical and operation skills required and establish MIC@Home as a distinct specialty [34]. Upskilling clinicians to handle diverse MIC@Home cases is also necessary. This includes integrating MIC@Home competencies into formal training programs, offering MIC@Home as an option for clinical rotation and short-term placements, ensuring continuous on-the-job training, and establishing a MIC@Home accreditation system were identified as strategic steps. These measures aim to prepare clinicians for the unique demands of MIC@Home and enhance patient care quality.

Furthermore, many participants noted persistent staff shortages during the initial implementation, and a consequent lack of dedicated MIC@Home care teams. Human resource policies could be adjusted to address this by reallocating clinicians to MIC@Home and assigning manual tasks to healthcare support or administrative staff to reduce the clinicians' workload.

### Future Steps.

Since various MIC@Home sites have different patient demographics, they will likely cater to specific patient profiles. As such, harmonization and coordination between MIC@Home sites will be beneficial, be it exchange of knowledge about best practices or resource sharing. Sharing best practices ensures that all MIC@Home sites maintain a high standard of care, regardless of patient demographics. This consistency provides clear guidelines for service delivery and ensures effective healthcare services across different locations. Additionally, collaboration fosters innovation as sites have a venue to share new ideas and solutions to common challenges.

With HaH now mainstreamed, there are more opportunities to partner with community care and social services providers for a more integrated healthcare system. These partnerships can alleviate the service burden on hospitals and provide a platform for post-discharge monitoring, rehabilitation, and social service aid, if required. Exploratory partnerships with private healthcare institutions could also be considered. Encouraging cross-sectoral collaboration and developing shared care models can enhance the scalability and effectiveness of HaH programs. However, a major consideration would be whether community care providers can attract and train qualitied staff, especially with existing labor shortages [35].

### Implications and next steps

As anticipated, managerial buy-in is essential to mainstream MIC@Home. Robust governance and regulatory oversight were also expected and observed to ensure patient safety and sustainable implementation. These findings align with previous research indicating the importance of organizational and regulatory structures in healthcare innovation [36].

Unexpectedly, the study revealed considerable resistance from healthcare providers about patient safety. This is despite HaH being prevalent overseas and published research. Additionally, the capacity of community care partnerships to alleviate hospital burdens was greater than anticipated, offering a promising avenue for future exploration.

This study has provided vital first-hand insights into the readiness of health system partners to implement and scale up the Mobile Inpatient Care at Home (MIC@Home) program in Singapore. By capturing the nuanced perspectives and experiences of key stakeholders through qualitative interviews, we have gained a deeper understanding of the factors influencing the adoption and scalability of this innovative care model. The views expressed by participants across different stakeholder groups have been meticulously summarized and collated.

This rich dataset forms a basis for subsequent research, which will employ policy lab discussions. These discussions aim to dive deeper into the specific implementation and policy strategies required to enhance the MIC@Home model's effectiveness and scalability. By engaging stakeholders, the research seeks to generate actionable insights to improve HaH services in Singapore, thereby informing policy-making and strategic planning processes in a more nuanced manner.

### Strengths and Limitations

This qualitative study has significant strengths. The method provides rich insights through thematic analysis. The inherent flexibility of semi-structured interviews allows researchers to dive into specific topics, and explore unforeseen areas of

interest. This adaptability is crucial for understanding the intricate dynamics of healthcare innovation implementation and scalability. Moreover, the flexibility to refine the interview guide after initial interviews helps to scope subsequent ideas that otherwise would have been unable to be discussed.

This study design also entails limitations. One of the challenges is the potential for researcher bias and the subjective nature of data interpretation, which could influence the conclusions drawn. Another limitation is the restricted generalizability of the findings. The insights obtained are context-specific, rooted in the experiences of a selected group of participants within Singapore's healthcare system. As such, while the study provides valuable insights for the implementation and scaling of HaH in Singapore, extrapolating these findings to different contexts requires caution.

## Conclusion

Expanding Hospital-at-Home care in Singapore involves navigating a complex landscape of technological, financial, regulatory, and cultural challenges. The insights from this study underscore the potential of HaH to transform healthcare delivery by making it more patient-centered, efficient, and sustainable. By addressing the gaps and leveraging the opportunities, Singapore can lead the way in scaling innovative care models that meet the evolving needs of its population. Future research should focus on fostering an enabling environment for HaH, ensuring that all stakeholders—patients, healthcare providers, policymakers, and technology partners—are aligned in their goals and efforts.

This study provides critical insights into the challenges and opportunities associated with scaling MIC@Home in Singapore, highlighting the importance of managerial support, governance, infrastructure, and stakeholder engagement. Despite encountering unexpected barriers, the findings underscore the potential of MIC@Home to enhance healthcare delivery. Future research should continue to explore the identified gaps, particularly focusing on technology adoption and community care partnerships, to refine and advance the implementation of MIC@Home.

## Supporting information

**S1 Material.** Theory of Change for MIC@Home.
(JPG)

**S2 Material.** Interview guide.
(DOCX)

**S3 Material.** Consolidated criteria for reporting qualitative studies (COREQ): 32-item checklist.
(PDF)

## Author contributions

**Conceptualization:** Crystal Min Siu Chua, Win Hon See Tho, Yuka Asada, Karen E Peters, Yi Feng Lai.

**Data curation:** Eward Wei Zheng Lim, Win Hon See Tho, Yi Feng Lai.

**Formal analysis:** Eward Wei Zheng Lim.

**Methodology:** Crystal Min Siu Chua, Eward Wei Zheng Lim, Win Hon See Tho, Yuka Asada, Karen E Peters, Yi Feng Lai.

**Supervision:** Crystal Min Siu Chua.

**Visualization:** Yi Feng Lai.

**Writing – original draft:** Eward Wei Zheng Lim, Win Hon See Tho.

**Writing – review & editing:** Crystal Min Siu Chua.

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
