## [Decision Letter · Decision Letter 0]

27 Nov 2024

PONE-D-24-50121Hospital-at-Home Care in Singapore: A Qualitative Exploration of Health System Partners’ State of Readiness, and Policy and Implementation Strategies Essential to Support Scale-UpPLOS ONE

Dear Dr. chua,

Thank you for submitting your manuscript to PLOS ONE. After careful consideration, we feel that it has merit but does not fully meet PLOS ONE’s publication criteria as it currently stands. Therefore, we invite you to submit a revised version of the manuscript that addresses the points raised during the review process.

We look forward to receiving your revised manuscript.

Kind regards,

Veincent Christian Pepito

Academic Editor

PLOS ONE

Journal Requirements:

“NO authors have competing interests”

Additional Editor Comments:

Dear authors, thanks for the manuscript. The manuscript shows promise but can further be improved. Here are my comments:

1. "Readiness" is such an abstract term, and there are many dimensions to readiness. Therefore, (a) please operationally define this phenomenon in the context of your study and (b) clarify how you will assess readiness qualitatively. On what indicators? On what grounds? On what aspects of readiness? It will probably help if you upload a copy of your data collection instrument so that we can see how you asked your questions.

2. Why did you not interview patients and service users? They would have given candid feedback on quality of care, readiness, and give recommendations for patient-centered care. Is there still a way for you to collect data from them to complete the picture for this study?

3. The study lacks a clear theoretical framework. There are many implementation research frameworks out there that could have been used to clarify the data analysis and reporting process. I strongly suggest that you employ one and use it in the analysis and reporting of your results to give it more structure and clarity.

4. The Results partly answer the objectives but it is less clear which parts of the results answer each objectives. Probably there is a need to signpost readers through appropriate headings and subheadings that these are the parts that answer these objectives on readiness and these parts answer the objectives on strategies. Probably employing the use of an appropriate implementation research framework can help clarify the analysis and the reporting of results.

5. The Discussion is good but along with the comment of one of the reviewers on liability, I want to ask you to expand on quality of care as a discussion point. How do we measure quality of care for hospital at home services? How do we ensure that care received at home is just as good (or even better) as quality of care in hospitals? Where does liability lie? We have written a brief commentary which could help you build your discussion on quality of care and liability: https://pmc.ncbi.nlm.nih.gov/articles/PMC10599638/

6. Please add a reflexivity and conditionality statement.

7. Please describe in more detail the MIC@home being evaluated by allotting a single subsection in the methodology explaining what this is and what is its theory of change.

8. The MIC@home, is it implemented by Singapore MOH and/or the National University of Singapore? If so, then the authors should be more transparent with their conflict-of-interest disclosure considering that the institutions writing this paper are the ones running the intervention and could be affected by the publication of the results of this study in this journal.

Reviewers' comments:

Reviewer's Responses to Questions

**Comments to the Author**

1. Is the manuscript technically sound, and do the data support the conclusions?

Reviewer #1: Partly

Reviewer #2: Yes

2. Has the statistical analysis been performed appropriately and rigorously? 

Reviewer #1: N/A

Reviewer #2: N/A

3. Have the authors made all data underlying the findings in their manuscript fully available?

Reviewer #1: Yes

Reviewer #2: Yes

4. Is the manuscript presented in an intelligible fashion and written in standard English?

Reviewer #1: No

Reviewer #2: Yes

5. Review Comments to the Author

Reviewer #1: - Needs more clarification on how the study participants were selected (more information about the sampling criteria.

- The authors need more explanation of the inclusion and exclusion criteria.

- Needs further clarification and discussion of the study theoretical framework and principles, specifically on HaH.

- The manuscript could benefit from language editing (language usage).

- (Figure 1) and (Figure 3a) must be revised.

Reviewer #2: I find the article to be well written and comprehensive. It provides valuable insight into the set-up of a new and novel service in healthcare.

As a clinician, the only gap that I found was regarding the ownership of liability when clinical incidents happen at home, as they invariably will. If the ministry or the institutions do not take ownership, then it will be a major barrier to healthcare professionals providing care at home. I am not sure if this has been addressed in any of the semi-structured interviews, but some mention can be made if it has.

6. PLOS authors have the option to publish the peer review history of their article (what does this mean? ). If published, this will include your full peer review and any attached files.

**Do you want your identity to be public for this peer review?** For information about this choice, including consent withdrawal, please see our Privacy Policy .

Reviewer #1: **Yes: ** Muhammad S. Al-Haddad

Reviewer #2: No

---

## [Author Response · Author response to Decision Letter 1]

2 Jan 2025

COMMENT BY EDITOR:1. Please ensure that your manuscript meets PLOS ONE's style requirements, including those for file naming.

RESPONSE TO EDITOR:Dear Editor, thank you for your comments. This manuscript is formatted based on PLOS One requirements.

COMMENT BY EDITOR:Thank you for stating the following in your Competing Interests section:

“NO authors have competing interests”. Please complete your Competing Interests on the online submission form to state any Competing Interests. If you have no competing interests, please state "The authors have declared that no competing interests exist.", as detailed online in our guide for authors at http://journals.plos.org/plosone/s/submit-now

RESPONSE TO EDITOR:Dear Editor, thank you for your feedback and help in making the changes. We have amended the cover letter to include the relevant information.

COMMENT BY EDITOR:PLOS requires an ORCID iD for the corresponding author in Editorial Manager on papers submitted after December 6th, 2016. Please ensure that you have an ORCID iD and that it is validated in Editorial Manager. To do this, go to ‘Update my Information’ (in the upper left-hand corner of the main menu), and click on the Fetch/Validate link next to the ORCID field. This will take you to the ORCID site and allow you to create a new iD or authenticate a pre-existing iD in Editorial Manager.

RESPONSE TO EDITOR:Dear Editor, thank you for your feedback. We have updated the ORCID iD.

COMMENT BY EDITOR:Your ethics statement should only appear in the Methods section of your manuscript. If your ethics statement is written in any section besides the Methods, please delete it from any other section.

RESPONSE TO EDITOR:Dear Editor, thank you for your comments. We have removed the ethics statement from other sections and only included it on page 9 in the manuscript.

COMMENT BY EDITOR:Dear authors, thanks for the manuscript. The manuscript shows promise but can further be improved. Here are my comments:

1. "Readiness" is such an abstract term, and there are many dimensions to readiness. Therefore, (a) please operationally define this phenomenon in the context of your study and (b) clarify how you will assess readiness qualitatively. On what indicators? On what grounds? On what aspects of readiness? It will probably help if you upload a copy of your data collection instrument so that we can see how you asked your questions.

RESPONSE TO EDITOR:Dear Editor, thank you for your insight. We agree that ‘readiness’ is an abstract term with multiple dimensions. We have added the definition on page 5. To assess readiness qualitatively, we relied on semi-structured interviews. Questions were based on prior literature with some of the following examples:- " In your opinion, are there any significant driving forces/motivators from you/your team/your organization for the scale-up to be successful?”

- Share your insights on the key system-level facilitators (to have worked well) and barriers to the scale-up of MIC@Home in Singapore. For clarity, we have attached a copy of our interview guide (data collection instrument) as supplementary material (S1 Material).

COMMENT BY EDITOR:2. Why did you not interview patients and service users? They would have given candid feedback on quality of care, readiness, and give recommendations for patient-centered care. Is there still a way for you to collect data from them to complete the picture for this study?

RESPONSE TO EDITOR:Dear Editor, we fully agree with your insights. Prior to this study, we have interviewed patients and caregivers on their perceptions and experiences:

1. Chua, C. M. S., Ko, S. Q., Lai, Y. F., Lim, Y. W., & Shorey, S. (2022). Perceptions of stakeholders toward “hospital at home” program in Singapore: a descriptive qualitative study. Journal of Patient Safety, 18(3), e606-e612.

2. Ko, S. Q., Chua, C. M. S., Koh, S. H., Lim, Y. W., & Shorey, S. (2023). Experiences of patients and their caregivers admitted to a hospital-at-home program in Singapore: a descriptive qualitative study. Journal of General Internal Medicine, 38(3), 691-698.

We have also provided references to these studies on pages 4 and 7. This study completes the picture by interviewing policymakers and organizational members, offering valuable insights into how to successfully implement the HaH model.

Additionally, this study's research questions are technical and require a deep understanding of the healthcare system and service delivery models. As most patients and service users lack this level of insight, we focused our sampling on the specified groups who could provide relevant expertise (page 8).

COMMENT BY EDITOR:3. The study lacks a clear theoretical framework. There are many implementation research frameworks out there that could have been used to clarify the data analysis and reporting process. I strongly suggest that you employ one and use it in the analysis and reporting of your results to give it more structure and clarity.

RESPONSE TO EDITOR:Dear Editor, thank you for your insights. The themes and subthemes described in our Results were developed inductively, and as such, they may not fully align with existing frameworks. However, we recognize that applying a framework could enhance structure and clarity.

To address this, we have integrated the Health Policy and Partnership Readiness Assessment Framework in our Introduction and Discussion section to better interpret and contextualize our findings (pages 2, 5 and 28-36).

We seek your kind understanding. Thank you once again for your valuable input.

COMMENT BY EDITOR:4. The Results partly answer the objectives but it is less clear which parts of the results answer each objectives. Probably there is a need to signpost readers through appropriate headings and subheadings that these are the parts that answer these objectives on readiness and these parts answer the objectives on strategies. Probably employing the use of an appropriate implementation research framework can help clarify the analysis and the reporting of results.

RESPONSE TO EDITOR:Dear Editor, thank you for your feedback. We agree that the Results can be clearer in answering the objectives and have added more in-text signposting for each of the themes on pages 11, 16 and 24.

COMMENT BY EDITOR:5. The Discussion is good but along with the comment of one of the reviewers on liability, I want to ask you to expand on quality of care as a discussion point. How do we measure quality of care for hospital at home services? How do we ensure that care received at home is just as good (or even better) as quality of care in hospitals? Where does liability lie? We have written a brief commentary which could help you build your discussion on quality of care and liability: https://pmc.ncbi.nlm.nih.gov/articles

/PMC10599638/

RESPONSE TO EDITOR: Dear Editor, thank you for your insights. We have added the reference on page 34.

We would also like to clarify that this study was not designed to assess quality, safety, equity, or liability issues, which have been well covered in the literature. For instance, two relevant Cochrane Reviews provide comprehensive analyses on these topics:

1. Edgar K, Ilić S, Doll HA, Clarke MJ, Gonçalves-Bradley DC, Wong E, et al. Admission avoidance hospital at home. Cochrane Database of Systematic Reviews 2024, Issue 3. Art. No.: CD007491.

2. Wallis JA, Shepperd S, Makela P, Han JX, Tripp EM, Gearon E, et al. Factors influencing the implementation of early discharge hospital at home and admission avoidance hospital at home: a qualitative evidence synthesis. Cochrane Database of Systematic Reviews 2024, Issue 3. Art. No.: CD014765.

With clinical effectiveness well represented in the literature, this study aims to highlight strategies that are necessary for the successful scaling of hospital at home programs.

Thank you for your understanding.

COMMENT BY EDITOR:6. Please add a reflexivity and conditionality statement.

RESPONSE TO EDITOR:Dear Editor, thank you for your suggestion. We have added reflexivity and conditionality statements on page 9.

COMMENT BY EDITOR:7. Please describe in more detail the MIC@home being evaluated by allotting a single subsection in the methodology explaining what this is and what is its theory of change.

RESPONSE TO EDITOR:Dear Editor, thank you for your comment. We have provided more context on MIC@Home with its theory of change under Settings (pages 7 and 8) and in S1 Material.

COMMENT BY EDITOR:8. The MIC@home, is it implemented by Singapore MOH and/or the National University of Singapore? If so, then the authors should be more transparent with their conflict-of-interest disclosure considering that the institutions writing this paper are the ones running the intervention and could be affected by the publication of the results of this study in this journal.

RESPONSE TO EDITOR:Dear Editor, thank you for your question. MIC@Home is implemented by the hospitals of Singapore. MOH Office for Healthcare Transformation and the National University of Singapore have no direct influence over implementation of MIC@Home. Therefore, we have declared no competing interest on page 40.

COMMENT BY REVIEWER 1:- Needs more clarification on how the study participants were selected (more information about the sampling criteria.

- The authors need more explanation of the inclusion and exclusion criteria.

RESPONSE TO REVIEWER 1:Dear Reviewer, thank you for your feedback.

The participants were purposefully sampled based on their insights to MIC@Home. Furthermore, this study's research questions are technical and require a deep understanding of the healthcare system and service delivery models. As most patients and service users lack this level of insight, we focused our sampling on the specified groups who could provide relevant expertise

We have added more explanation to the patient eligibility criteria on pages 6 and 7.

COMMENT BY REVIEWER 1:- Needs further clarification and discussion of the study theoretical framework and principles, specifically on HaH.

RESPONSE TO REVIEWER 1:Dear Reviewer, thank you for your insight, we have given our Discussion more structure with references to the Readiness Assessment Framework. This can be found on pages 2, 5 and 28-36.

COMMENT BY REVIEWER 1: - The manuscript could benefit from language editing (language usage).

RESPONSE TO REVIEWER 1:Dear Reviewer, thank you for your feedback. We have reviewed our manuscript and completed language editing.

COMMENT BY REVIEWER 1:- (Figure 1) and (Figure 3a) must be revised.

RESPONSE TO REVIEWER 1:Thank you for your feedback. Figure 1 and Figure 3a represent detailed themes and subthemes that were inductively derived from our data. Instead of revising these figures, we have now incorporated the Readiness Assessment Framework to guide the Discussion section based on our results. This approach will provide greater clarity and highlight actionable insights moving forward.

We kindly seek your understanding and appreciate your valuable input.

COMMENT BY REVIEWER 2:Reviewer #2: I find the article to be well written and comprehensive. It provides valuable insight into the set-up of a new and novel service in healthcare.

RESPONSE TO REVIEWER 2:Dear Reviewer, thank you for your kind encouragement.

COMMENT BY REVIEWER 2:As a clinician, the only gap that I found was regarding the ownership of liability when clinical incidents happen at home, as they invariably will. If the ministry or the institutions do not take ownership, then it will be a major barrier to healthcare professionals providing care at home. I am not sure if this has been addressed in any of the semi-structured interviews, but some mention can be made if it has.

RESPONSE TO REVIEWER 2:Dear Reviewer, thank you for your insight, we fully agree with you that patient safety is important.

We would like to stress that hospitals remain liable for MIC@Home’s patient clinical outcomes, and that supporting policies, regulations and guidelines are crucial for ensuring quality and safety of patient care (pages 33, 35-36).

---

## [Decision Letter · Decision Letter 1]

25 Feb 2025

PONE-D-24-50121R1Hospital-at-Home Care in Singapore: A Qualitative Exploration of Health System Partners’ State of Readiness, and Policy and Implementation Strategies Essential to Support Scale-UpPLOS ONE

Dear Dr. chua,

Thank you for submitting your manuscript to PLOS ONE. After careful consideration, we feel that it has merit but does not fully meet PLOS ONE’s publication criteria as it currently stands. Therefore, we invite you to submit a revised version of the manuscript that addresses the points raised during the review process.

We look forward to receiving your revised manuscript.

Kind regards,

Veincent Christian Pepito

Academic Editor

PLOS ONE

Additional Editor Comments:

Dear authors,

Thanks for your revision. The reviewers have advised acceptance, but I still have a few comments:

1. The abstract needs to be re-written. The framework used should be in the Background/Methods part, and not in the Results. This should be fixed.

2. A brief description of the framework used should be put in the Introduction, also explaining why that particular framework was used. Also, please improve the integration of the framework with the rest of the article. As it stands, the framework does not appear to be well-integrated with the methods and results. The framework should in some sense, guide the questions asked, the analysis, and the presentation of the results.

Reviewers' comments:

Reviewer's Responses to Questions

**Comments to the Author**

1. If the authors have adequately addressed your comments raised in a previous round of review and you feel that this manuscript is now acceptable for publication, you may indicate that here to bypass the “Comments to the Author” section, enter your conflict of interest statement in the “Confidential to Editor” section, and submit your "Accept" recommendation.

Reviewer #1: All comments have been addressed

2. Is the manuscript technically sound, and do the data support the conclusions?

Reviewer #1: Yes

3. Has the statistical analysis been performed appropriately and rigorously? 

Reviewer #1: Yes

4. Have the authors made all data underlying the findings in their manuscript fully available?

Reviewer #1: Yes

5. Is the manuscript presented in an intelligible fashion and written in standard English?

Reviewer #1: Yes

6. Review Comments to the Author

Reviewer #1: (No Response)

7. PLOS authors have the option to publish the peer review history of their article (what does this mean? ). If published, this will include your full peer review and any attached files.

**Do you want your identity to be public for this peer review?** For information about this choice, including consent withdrawal, please see our Privacy Policy .

Reviewer #1: **Yes: ** Muhammad S. Al-Haddad

---

## [Author Response · Author response to Decision Letter 2]

8 Apr 2025

All relevant data are within the manuscript and its Supporting Information files as Supplementary Files 1, 2, and 3. We would like to confirm that our minimal data set is present. I confirm that the manuscript and supporting files (S1 Material.jpg, S2 Interview Guide.docx, and S3 Material.pdf) include the minimal data set as defined by PLOS ONE. No further data uploads are required. I mistakenly selected the assistance checkbox at submission, but do not require additional support.

Reviewer comments: Dear authors, Thanks for your revision. The reviewers have advised acceptance, but I still have a few comments: 1. The abstract needs to be re-written. The framework used should be in the Background/Methods part, and not in the Results. This should be fixed.

Response to reviewer: Dear Editor, thank you for reviewing our manuscript, and your comments. We have reorganized the abstract accordingly on page 2.

Reviewer comments: 2. A brief description of the framework used should be put in the Introduction, also explaining why that particular framework was used. Also, please improve the integration of the framework with the rest of the article. As it stands, the framework does not appear to be well-integrated with the methods and results. The framework should in some sense, guide the questions asked, the analysis, and the presentation of the results.

Response to reviewer: Dear Editor, thank you for your feedback. We have added some context about the framework on pages 5 and 6. Furthermore, we have integrated the framework more thoroughly within the Methods section (page 9) and the Results section (pages 12, 13, 17, and 27). This ensures consistency with its use in guiding the Discussion segment, as previously outlined.

With this addition, the framework now would have been integrated to guide our research questions, analysis in methods, and presentation of results. Thank you.

---

## [Editor Report · Decision Letter 2]

14 Apr 2025

Hospital-at-Home Care in Singapore: A Qualitative Exploration of Health System Partners’ State of Readiness, and Policy and Implementation Strategies Essential to Support Scale-Up

PONE-D-24-50121R2

Dear Dr. chua,

We’re pleased to inform you that your manuscript has been judged scientifically suitable for publication and will be formally accepted for publication once it meets all outstanding technical requirements.

Kind regards,

Veincent Christian Pepito

Academic Editor

PLOS ONE
---

## [Editor Report · Acceptance letter]

PONE-D-24-50121R2

PLOS ONE

Dear Dr. Chua,

I'm pleased to inform you that your manuscript has been deemed suitable for publication in PLOS ONE. Congratulations! Your manuscript is now being handed over to our production team.

Kind regards,

on behalf of

Mr Veincent Christian Pepito

Academic Editor

PLOS ONE